# Scale-up of Electrospinning: Market Overview of Products and Devices for Pharmaceutical and Biomedical Purposes

**DOI:** 10.3390/pharmaceutics13020286

**Published:** 2021-02-22

**Authors:** Safaa Omer, László Forgách, Romána Zelkó, István Sebe

**Affiliations:** 1University Pharmacy Department of Pharmacy Administration, Semmelweis University, Hőgyes Endre Street 7-9, 1092 Budapest, Hungary; safaa.omer@phd.semmelweis.hu; 2Department of Biophysics and Radiation Biology, Semmelweis University, Tűzoltó Street 37-47, 1094 Budapest, Hungary; forgach.laszlo@med.semmelweis-univ.hu

**Keywords:** electrospinning (ES), nanofibers (NFs), scale-up, pharmaceutical industry

## Abstract

Recently, the electrospinning (ES) process has been extensively studied due to its potential applications in various fields, particularly pharmaceutical and biomedical purposes. The production rate using typical ES technology is usually around 0.01–1 g/h, which is lower than pharmaceutical industry production requirements. Therefore, different companies have worked to develop electrospinning equipment, technological solutions, and electrospun materials into large-scale production. Different approaches have been explored to scale-up the production mainly by increasing the nanofiber jet through multiple needles, free-surface technologies, and hybrid methods that use an additional energy source. Among them, needleless and centrifugal methods have gained the most attention and applications. Besides, the production rate reached (450 g/h in some cases) makes these methods feasible in the pharmaceutical industry. The present study overviews and compares the most recent ES approaches successfully developed for nanofibers’ large-scale production and accompanying challenges with some examples of applied approaches in drug delivery systems. Besides, various types of commercial products and devices released to the markets have been mentioned.

## 1. Introduction

Ideal drug delivery systems have been developed to achieve the best therapeutic effects and lowest toxicity problems [1]. The rapid progress in the field of nanotechnology has led to the development of many techniques for the production of numerous nano-scale composites [2,3], of which nanofibers have received considerable attention [4] due to diversity in the fabrication technologies and applications [5,6], especially in the fields of pharmaceutical drug delivery [7,8,9], biomedical applications including wound dressing [10,11,12,13], and tissue engineering [14,15]. Electrospinning (ES) has been considered one of the most efficient techniques used for the synthesis of nanomaterials since the 20th century [16], and great works have been done in the late 1990s and early part of the 21st century [17,18]. ES is a simple, highly efficient and reproducible process [19]. It depends on the application of a high electrical field between a metallic needle (containing the polymeric solution or melt) [20] and a grounded collector [21,22]. Above a certain critical value [23], the liquid jet is ejected from the tip of a needle, forming a Taylor cone [24], followed by subsequent elongation, thinning, and deposition of sub-micron fibers on the collector [2,25]. Various polymers have been used in the production of nanofibers (NFs) [26,27] with unique properties [28,29,30], which make NFs a suitable candidate for many pharmaceutical applications, such as solubility improvement [31,32,33,34,35,36], oral delivery of biopharmaceutical [37], controlling the drug release [38,39,40], as well as one-step co-encapsulation of one or more drugs [41,42]. Various spinnerets and collecting-electrode devices and accessories are used [43] to produce different fiber mats, including non-woven fiber mats, aligned fiber mats, patterned fiber mats, random three-dimensional structures, and sub-micron spring and convoluted fibers [44,45]. Sometimes multi-axial capillaries are exploited to produce NF mats that serve different application purposes [46,47,48], or to enable the fabrication of nanofibers from “unspinnable” liquids [49], or to enhance the quality of resulting nanofiber structures [50]. This means that the final quality of NFs is governed by solution properties, process used, and environmental conditions [51]. Since ES has become a novel, versatile, and applicable technology in the pharmaceutical research and drug development process, switching to industrial manipulation is of critical importance through scaling-up the production, because the laboratory-scale electrospinning device with a single needle has a rather low (0.01–2 g/h) productivity [52]. Even though the ES technique is relatively simple and easy to handle, it faces many challenges that need to be managed in order to have fibers of a high quality [53], such as the inconsistent properties between different batches when using natural polymers such as silk fibroin [54], the toxicity problem resulting from a residual organic solvent, and the stability of active agents with thermal treatment [55]. Moreover, it is a relatively slow and time-consuming process, the quality of the final NFs is affected by the electrical field interference when using multiple needles to increase the production, the improper optimization of parameters will adversely affect or even interfere with fiber formation [56], and difficulty in the NF fabrication of native materials of plant origin, such as starch, necessitates some modifications that adversely interfere with the properties of native materials [57]. Different attempts to use needle-free technologies have been carried out to overcome the limitations, but further studies are needed to formulate NFs with improved quality that fit the pharmaceutical industry [56]. Although the reviews that demonstrate the scale-up of ES technology have been rapidly growing recently, the majority of studies and reviews available mainly focused on the principle of ES, its applications, parameter optimization, and characterization; therefore, light must be shed on the different approaches that have been proved or exploited for massive NF production, particularly for pharmaceutical drug delivery. This review aims to give an overview of the ES technologies developed for the fiber mass production, the challenges encountered, and light touches to products and devices that have been or are near to being released to the markets, emphasizing pharmaceutical-industry-relevant examples.

## 2. Up-Scaling of Nanofiber Production

Many approaches have been described in the literature for increasing nanofiber productivity. They focus on increasing jet numbers through single-needle modification, using multiple needles, and needleless electrospinning [58,59,60,61]. Force-spinning, as a newly developed method which utilizes either centrifugal forces instead of electrostatic force [42] or a combination of both forces, has been found to give a profound effect [43]. Up to date, free-surface and centrifugal methods have been explored for their potential production volume up-scaling [62].

### 2.1. Challenges Facing the Scaling-up Process

Although it is easy to produce NFs in the lab, when it comes to industrial-scale production, mass productivity always remains a significant concern together with a collection and downstream processes, and there are many challenges which act as an obstacle for nanofiber production in the pharmaceutical industry [63]. The large-volume processing is a time-consuming process that will not fit the pharmaceutical industry capacity. Furthermore, safety and environmental attributes arise from organic solvent for scaled-up electrospinning [62]. The difficulty of process optimization is that only some of the electrospinning process parameters can be easily varied [63,64], which is still a significant challenge. Needleless technology suffers the fast evaporation of volatile solvents used for the solubilization of poorly water-soluble drugs; therefore, the concentration will change and make the process harder to handle, although many attempts have been made to overcome this problem. It is still a challenge, especially with highly concentrated solutions and highly volatile solvents, which leads to decreasing accuracy and reproducibility in all fabrication stages [14]. When using multi-needle spinnerets, needles clogging and the interference of close by needles will inversely impact the jet formation and final quality of the fiber mats. The improper configuration of collectors and downwards spinning of the jets result in beads or even interfere with fiber formation [56]. It is well known that natural polymers have gained increasing interest in pharmacy, but, unfortunately, it is challenging to fabricate NFs from starch and similar natural polymer from pure materials because they have no sufficient mechanical strength beside their relatively low thermostability; this means that without modification it will be impossible, taking into consideration that the modification processes may negatively impact the function of natural polymers [57]. Inconsistent properties between different batches when using natural polymers such as silk fibroin may occur. Besides, silk fibroin plays a role in stabilizing the pharmaceutical agents, and it needs special treatment with organic solvents to increase its stability. However, using organic and caustic solvents might damage the structure and bio-activity of the bio-molecules and susceptible drugs [54].

### 2.2. Increasing Jet Production from a Single Needle

Multi-jet electrospinning was first accomplished using a single nozzle with a grooved tip, from which branches of multi-jets were formed, which in turn increases productivity from a single needle. Beadless membranes of poly-butadiene (PB) were successfully prepared by this method. Nevertheless, there was no clear cut about nanofibers’ mass production of a single needle. The theory behind this method was based on the fact that increasing jet production might be attributed to the increased voltage, which causes inconsistent current distribution, and clogging of the passage-way of the polymer solution [65]. Jet splitting has been actually observed when fluid jets interact with electric fields, so it is not a just postulation [66]. Different cross-sectional shapes have been observed from a polymeric solution for fiber fabrication by electrospinning from which fibers that were split longitudinally from larger fibers were observed [67]. Only a few studies have been carried out because it is not as efficient as other methods.

### 2.3. Increasing Jet Production through Multi-Needle Electrospinning

A straightforward and easy way of mass producing nanofibers is by utilizing a multiple-needle instead of a single needle called “multi-needle electrospinning.” [17,68,69,70]. Blended biodegradable nanofiber mats with different weight ratios of poly-vinyl alcohol (PVA) and cellulose acetate (CA) were successfully fabricated via multi-jet electrospinning; the results showed that the blended nanofiber mats have good dispersibility [54]. The process increases the output and enables simultaneous electrospinning of different materials along with the possibility of controlling and maintaining the NFs properties [71]. Nevertheless, this technique faces two main problems: electrostatic field interaction between needles and needles clogging [58,72], which will adversely impact the quality of the produced nanofibers. Therefore, a process modification and developments are of paramount importance. No notable increase in production rate has been observed [14]. Many attempts to overcome these problems have been carried out, including: using a large operating space and careful design of needles to address the problem of charge repulsion and uneven fiber deposition [73]. Therefore, different arrangements may be applied to the needles such as one-dimensional linear configuration or two-dimensional configurations such as elliptical, circular, triangular, square and hexagonal [65,74]. An arced multi-nozzle spinneret was designed to increase the production efficiency of electrospinning, resulting in relatively uniform electrical fields which in turn decreases the electrical interference of the needles and subsequently improves nanofiber quality [75]. The following are some examples for applying this technology for drug delivery and the pharmaceutical industry: an electro-spun formulation containing galactosidase as a model biopharmaceutical drug has been successfully developed through using scaled-up electrospinning experiments of a lab-scale, high-speed electrostatic spinning (HSES) setup consisting of a circular-shaped, stainless steel spinneret, equipped with eight orifices, and connected to a high-speed motor. The feeding rate was increased to approximately 30 times higher than the usual feeding rate for single-needle electrospinning. The results showed that the content and the added excipients enabled appropriate grinding of the fibrous sample without secondary drying, and subsequently the preparation of pharmaceutical tablets. According to the obtained results, high-speed electrospinning is a viable alternative to traditional biopharmaceutical drying methods, especially for heat-sensitive molecules, and tablet formulation is achievable from the electrospun material prepared in this way [52].

Another interesting example is the formulation of a new intravenous (i.v.) bolus dosage form of doxycycline (DOX) by using a high-speed electrospinning (HSES) setup consisting of a stainless-steel spinneret equipped with 36 orifices and connected to a high-speed motor. The produced fibrous material was collected by a cyclone with a high (~80 g/h) productivity rate. The freeze-dried product was also prepared from the same precursor solution of HSES for comparison. The technology produced an amorphous, uniformly distributed DOX fibrous powder. The result of dissolution showed that the produced fibrous powder has a seven times higher dissolution rate than that of the marketed formulation, which confirmed that the reconstitution solution could be applied as an i.v. bolus dosage form. Subsequently, this work confirmed that the continuous high-speed electrospinning process could be a viable high productivity alternative to batch freeze-drying process [34]. In light of the aforementioned examples, it can be concluded that scaled-up multiple nozzle electrospinning technologies can be successfully used in the pharmaceutical industry after adjusting the process parameters, modification of spinneret by utilizing orifices or holes instead of needles, and through increasing the speed of spinning as shown in the two examples; thus, these configurations fulfil the requirements of the pharmaceutical industry.

### 2.4. Nozzleless (Free-Surface) Technologies

In this technology, the NF production is directly from an open surface instead of needles. The basic principle behind the formation of multi-jet from this system is as follows: the wave of an electrically conductive liquid is self-organized, followed by simultaneous multiple-jet formation when the applied voltage is above the critical value [65]. In contrast to multiple needles-based electrospinning, free-surface electrospinning requires a very high voltage to overcome the liquid’s surface tension, which depends on the type of spinneret used. It is also difficult to maintain consistent solution viscosity owing to solvent evaporation from the surface [76]. However, the technology can be used successfully to address the clogging problem associated with the multiple-needle technique because the fiber formation is obtained without the need for needles [14]. Spinnerets for needleless electrospinning are classified into two categories: rotating and stationary spinnerets. The rotating spinnerets are capable of generating a mechanical rotation to the polymeric solution, while the stationary spinnerets require an auxiliary force to initiate the process (e.g., magnetic field, gravity, and gas bubble) [69].

#### 2.4.1. Electrospinning Method Utilizing Rotating Spinneret

Many rotating spinnerets are available, from which cylinder, disc, ball, and wire are mostly reported [56]. The spinnerets are connected with a high voltage power supply and immersed in the spinning solution. The disc spinneret needs a relatively lower applied voltage than the cylindrical spinneret to initiate fiber formation, and the fibers were mainly formed on the top disc. Both electrospinning systems (disc and cylinder) could produce uniform nanofibers, but the fibers will be thinner with the disc than with the cylinder at the same process conditions [56]. The electrospun NFs are collected from an upward direction, which ensures drop-free fibers [69]. These types of spinnerets have superior advantages over needle-based electrospinning. They have production rates of 8.6 g/h with a cylinder, 6.2 g/h with a disk and 3.1 g/h with a ball as the spinneret for polyvinyl alcohol (PVA) fibers under the same working conditions [69,77]. The formation of multiple jets utilizing a charged cylindrical electrode from the surface of the polymeric solution was the first patent regarding this technology [78], followed by the development of the world’s first industrial free-surface electrospinning setup (Nanospider^®^) by the Elmarco Company (Liberec, Czech Republic) with nanofibrous non-woven membranes obtained with a 50–500 nm diameter at a production rate of 1.5 g/min per meter of roller length [71,79]. The technology is widely used to manufacture NFs for pharmaceutical applications; a notable example of this is the investigation of electrospun fibers containing antiretroviral drugs tenofovir (TFV) as a new antiviral topical formulation against HIV-1. Manufacturing scalability of drug-eluting fibers was assessed using a NanospiderTM NS-1WS500U large-scale production instrument (Elmarco, Inc., Liberec, Czech Republic). NS-1WS500U (wire instrument) and the parameters were optimized for the best fibers production. The TFV loading in fibers increased with increasing the TFV per cent in solution, and encapsulation efficiency was improved. The study results have specified the important parameters for the scale-up production of TFV drug-eluting fibers by wire electrospinning and support the possibility of the scale-up and transferability of TFV-loaded electrospun fibers to the pharmaceutical industry [80].

#### 2.4.2. Electrospinning Method Utilizing a Stationary Spinneret

In contrast to rotating spinnerets when using stationary spinnerets, an external source of force is required for jet initiation such as a magnetic force [81], high-pressure gas flow [65,82], gravity and ultrasound radiation [83]. Again, similar to rotating spinnerets, this method produces nanofibers with higher production rates than multiple-needle ES. The process setup is more complicated, and the produced nanofibers are relatively coarse [69]. The first study produced nanofibers from dextran utilizing a steel rod and collector as a needleless electrospinning setup to spin solutions directly from the surface. After parameter optimization, the rod diameter is an important parameter for the Taylor cone numbers and electrospinning productivity. The results showed a spinning performance of 0.67 g/min per meter and an average diameter of 162 nm. This study set the parameters necessary for high-quality NFs of dextran fabrication using needleless technology to be used for drug delivery [83]. Another example is a fabrication of antimicrobial polycaprolactone (PCL) NFs using uncommon needleless and collectorless alternating current (AC) electrospinning. The morphological evaluation of resultant fibers showed smooth beadless NFs [84].

Recently, many theoretical and practical studies have been carried out to develop the free-surface method for large-scale production [85]. A novel modified spinneret based on needleless technology called mushroom has been developed to increase the quality and yield of NF production by generating stable pre-Taylor cones with high curvature, resulting in the production rate of 13.7 g/h [86]. To increase the electric current, an auxiliary electrode can be introduced [17,87]. For instance, a modified spinneret consisting of a narrow, long gutter bounded by a metal electrode having a sharp edge has been developed for a continuous high production rate of NFs. The concept behind mass production is explained by the highest electrical charge density that forms along the sharp edge, which will lead to many self-assembled Taylor cones. The technology was assessed using polyacrylonitrile/dimethyl-formamide and polyvinylpyrrolidone/ethanol solutions. The small size of the prototype made possible a 20–50 times increase in productivity compared to the single capillary method [88].

### 2.5. Other Approaches

Significant improvement in production was obtained by introducing the polymer melt differential electrospinning (PMDES) method, which enables the production of multiple jets with the smallest inter-jet distance of an umbrella-shaped spinneret addition. Suction wind and multistage electric field were proposed to refine the fibers [70]. A centrifuge spinning method that utilizes centrifugal force, rather than the electrostatic force can be successfully used for solution and melt electrospinning. The device consists of a multiple orifice spinneret, thermal system, collecting devices, environmental chamber, control system, motor, and brake. Some researchers have utilized centrifuge spinning to fabricate nanofibers from several ordinary materials [59], other setups based on utilizing air-sealed centrifugal spinning [89]. Alternating current electrospinning (ACES) was also compared to direct current electrospinning. The results showed comparable nanofiber properties with the capability of ACES to increase the productivity of the mat by approximately six-folds [90]. Hybrid electrospinning methods that used secondary assistant force for the mass production of NFs have also been studied [59]. A new approach for the mass production of NFs is based on the combination of magnetic and electric fields acting on a two-layer system [81].

Another technology has been proven for the production of co-axial NFs on a large scale. The technology is based on the formation of bilayer liquid on the surface of needleless weir spinneret, which eventually supports co-axial nanofibers’ formation. The results demonstrated an increased production rate of uniform core/shell NFs compared to needle coaxial spinneret [91]. A comparative study of traditional electrospinning (TES) with ultrasound enhanced electrospinning (USES) has been conducted for the formulation of a nanofiber drug delivery system using polyethylene oxide (PEO) and chitosan as carrier polymers and theophylline anhydrate as a water-soluble model drug. The results showed that NFs produced by USES were amorphous compared to those produced by TES [92]. A very promising novel application of the free-surface method has been manipulated by combining the alternating current electrospinning (ACES)—instead of using direct current—with centrifugal force, and a higher throughput has been obtained [93]. Another study involves utilizing centrifugal and electrostatic forces simultaneously for the fabrication of NFs from poly-acrylonitrile (PAN) and poly-L-lactic acid (PLLA) [94]. Modified free-surface electrospinning (MFSE) using a cone-shaped air, and added sodium dodecylbenzene sulfonates (SDBS) to generate bubbles on the liquid surface. The results revealed that the quality and production rate were increased [95]. A very promising study, compatible with the pharmaceutical industry has been developed for the fabrication of a poorly water-soluble drug itraconazole utilizing scalable, high-speed electrospinning setup. The scaled-up experiment was carried out using a stainless steel spinneret with sharp edges and spherical cap geometry connected to a high-speed motor. In addition to high production output (75-fold productivity improvement, i.e., 450 g/h), the obtained morphology and quality were similar to that produced by single-needle electrospinning. Therefore, it could be possible to have a continuous, efficient, and scaled-up method that meets the pharmaceutical industry’s requirements [33].

## 3. Impact of Scaling Condition on the Nanofibers’ Functionality-Related Properties

The morphology of nanofibers is controlled by electrospinning process parameters (flow rate, viscosity, the distance between tip and collector, solvent conductivity, temperature, and humidity); so far, it is not an easy process to have nanofibers with desired properties and architecture, because it is still challenging even for conventional electrospinning [96]. Increasing the flow rate to increase the throughput will affect the morphology of electrospun fibers and leads to the formation of beads due to incomplete drying results in increased pore size and fiber diameters [2]. The collector types and configurations have also been demonstrated to affect the final NF properties, for example, polycaprolactone (PCL) fibers were produced by electrospinning using three collectors: rotating drum static copper wires, and a rotating mandrel and the effect on morphology was evaluated. The best fiber alignment and lowest average fiber diameter were obtained using parallel copper wires with a 1 cm gap [97]. There are many published works comparing the properties of nanofibers fabricated through scaled-up and conventional needle electrospinning methods. Free-surface electrospinning (wire electrode) for the high-throughput fabrication of fibers delivering tenofovir (TFV) was compared to the conventional needle electrospinning method (needle electrode), polyvinyl alcohol (PVA) fibers containing up to 60% TFV were fabricated, and all electrospinning solutions were in the same conditions with regard to solution conductivity, viscosity, surface tension, or pH. The resultant fibers were evaluated for physicochemical parameters such as fiber morphology, drug crystallinity, drug loading and release kinetics. The results showed that electrospinning using needle and wire instruments produced materials with similar mesh and fiber properties, and fiber diameter decreased with increasing drug loading for both the needle and wire instruments, except for formulations used to fabricate 60% TFV fibers, in which the wire instrument produced fibers with slightly smaller diameters compared to the needle instrument, which was attributed to greater solution conductivity with increasing drug loading and the greater overall electric field strength of the wire. Additionally, the electrospinning of both instruments showed that the finished fibers have a generally smooth and cylindrical morphology. The actual drug loading and encapsulation efficiency of TFV was comparable between the needle and wire electrospinning; the only significant difference observed between actual drug loading and encapsulation efficiency values was for the 60% TFV fibers in which the fibers produced by the wire instrument had a 10% decrease in absolute actual drug loading compared with the fibers produced by the needle instrument. This difference was attributed to the settling of drug precipitate in the carriage tube during electrospinning with the wire instrument. To overcome the problem, a more uniform micronization of the drug before electrospinning was suggested or actively mixing the polymer solution in the reservoir during electrospinning. It can be concluded that the TFV–PVA fibers were successfully transferred from a laboratory-scale to a production-scale instrument [80]. Another study utilizing high-speed electrospinning (HSES) that fit the pharmaceutical requirements was used to prove that a higher production rate will be obtained than that of a single-needle electrospinning (SNES) setup. Itraconazole (ITRA) electrosspun nanofibrous material was formulated using HSES, SNES, spray drying (SD), and film casting (FC). The nanofiber products were evaluated in terms of the dissolution, morphology, and amorphous dispersion. The results showed fast dissolution with both scaled-up and single-needle electrospun fibers. The obtained morphological properties were the same for both lab-scale SNES and the scaled-up HSES method with a diameter range of 0.5–2 mm. Further studies are expected to explain the reason behind some beads’ micrometer size in the case of HSES nanofibers [33].

## 4. Development of Electrospinning Machines from Laboratory to Industrial Scale

The continuous increase in passion and research in electrospinning has led to increased competition among laboratory-scale equipment’s suppliers. The market movement was revived with various spinning and collecting-electrode devices and accessories. For example, 4Spin Company (Dolní Dobrouč, Czech Republic) offers highly modular systems and polysaccharides such as hyaluronic acid, chitosan, or cellulose can spin, resulting in a fiber diameter of 300–500 nm [98]. E-Spintronic equipment (Gernlinden, Germany) is characterized by being easy to clean and capable of forming 3D spinning movement with a speed of 1–600 mm/min [99]; information on the fiber diameter range is not available on their website. HOLMARC Opto-Mechatronics (Kerala, India) designed equipment suitable for the production of protein NFs, carbon nanotubes, and inorganic NFs of diameter size 50–5000 nm. They provide systems for research in various fields, such as thin films, biotechnology, nanotechnology, life sciences [100]. NEU KatoTech Co. Ltd. device (Kyoto, Japan) uses electrospinning techniques to safely and easily produce NFs with diameters of 50 to 800 nm. This device is widely used in the automotive industry to research and develop filters and fuel cells [101]. Nadetech Innovations (Navarra, Spain) produce completely automatized ES systems, with completely controllable ES parameters with high accuracy and reproducibility. Their system uses single, coaxial, tri-axial or multi-nozzle spinnerets [102]. Physics Equipments (Chennai, India)—similar to the E-Spintronic equipment—offers devices with a spinning chamber constructed of aluminium strut frames, two polycarbonate clear doors, a window and a fiberglass panel with a voltage sensor, emergency stop and multiple safety features (a core-shell nozzle spinneret is also available); fiber diameter data are not available [103]. A spinbox instrument by Bioinicia (Valencia, Spain) designed for the lab-scale fabrication of micro- or nano-structured fibers and particles for use in a wide variety of applications including regenerative medicine, drug delivery, micro-encapsulation (of food and skin-care ingredients), functional textiles, and filtration [104]. Besides, various basic, intermediate and advanced kits are available, engineered by Bioinicia and Fluidnatek systems, for research purposes; their large-volume solution feeding system for extended production batches is remarkable. Their product line offers models with easy-to-clean construction, which are dedicated to Cleanroom-ES in the biomedical and pharmaceutical field [105]. Spraybase Company (Kildare, Ireland) supplies different types of spinnerets, including coaxial and tri-axial. Extraordinarily, a melt ES device is available in their line-up [106].

Therefore, a wide range of laboratory-scale equipment is available in the market. The majority is based on needle-type electrospinning, with a low production rate, but is compatible with research capacity. In recent years, many companies have attempted to address low productivity by developing new methods adapted from conventional electrospinning to increase production. Thereafter, the potential for the use of NFs in industrial applications has become feasible [107]. Many companies have released electrospinning equipment for industrial production to the markets, which could be used for pharmaceutical applications (Table 1). Many of them use nozzle-based technologies to control the NF properties, particularly when they are exploited as drug delivery products, while others are based on needleless electrospinning (NLES). Unfortunately, no large number of devices applicable to pharmaceutical industries have been released to date. Nevertheless, the research works are increasing daily based on NLES technology in an attempt to find novel techniques and instruments compatible with the pharmaceutical industry [56], for example, Bioinicia (Valencia, Spain) [105], Elmarco [108], and Fnm Co. (Fanavaran Nano-Meghyas) (Baghestan, Iran) [109]. The first patent in nanofiber scale-up production is based on the possibility to create Taylor cones and the subsequent flow of material not only from the tip of a capillary, but also from a thin film of a polymer solution [78], which led to the development of the world’s first industrial free-surface electrospinning setup (Nanospider^®^) by the Elmarco Company (Liberec, Czech Republic). It is based on needle-free electrospinning technology and designed for the effective production of the highest-quality nanofibers with low solvent consumption, continuous production, and a wide range of solvent for polymer solubility. Furthermore, it is a versatile technology and is easily adapted to various process parameters to optimize the specific properties. The Nanospider™ NS 8S1600U is the base unit for industrial production. It is characterized by being scalable to up to four units and designed for 24 h/7 days operation. Throughput depends on the polymer, substrate, process and fiber diameter, for example: 20,000,000 m^2^/year for PA6 on cellulose, nanofiber layer width: 1.6 m, basis weight: 0.03 g/m^2^, fiber diameter: 150 nm +/− 30% [108].

Bioinicia (Valencia, Spain) is the first company to have overcome the up-scaling issues related to the electro-hydrodynamic-spinning and electro-spraying technology with a large scale; it has reached a unique positioning in the pharmaceutical industry, being the only company fully capable of developing and producing drugs with optimized performance. The instrument is versatile with continuous ISO and GMP-certified production of pharmaceutical products: one can use any solvent, excipient or API (even live cells) at solutions, suspensions, and emulsions; no stress is applied to the API at the highest product stability and quality; dry product in a single step (easy downstream process); minimal moisture content and excellent processability of the final product with high reproducibility and dose uniformity; narrow and controllable particle size distribution (50 nm–10 µm) [105]. Similarly, electrospinning (Tong Li Tech)/NaBond (Hong Kong) has developed laboratory-, pilot-, and industrial-scale equipment; a wide range of instrumental selection is available, including a dedicated bio-medical instrument, and various techniques can be applied, such as needle-based electrospinning, near-field electrospinning, coaxial electrospinning, 3D print electrospinning, melt electrospinning, and continuous electrospinning [110]. Another commercialized instrument that can be used in the pharmaceutical industry is PE3550 from INOVENSO (İstanbul, Turkey); initially conceived for the production of air filtration nanofiber-based products (N95/N99 filtration media), it enables high-throughput NF production based on needles and core-shell electrospinning. PE3550 is equipped with 168 electrospinning nozzles, allowing the surface to be uniform with a fiber diameter of 50–400 nm, and it also enables continuous production at the required quality even under changing environmental conditions [111]. Other companies that supply electrospinning equipment for both laboratory- and large-scale production include LINARI NanoTech (Pisa, Italy), offering coaxial needle/multi-needle systems, which use up to eight independently controlled syringe pumps and have an automatic cleaning system in addition to internal temperature and humidity control [112]. Nanoflux (Singapore) is a multi-nozzle system for the continuous production of the Nanofiber fabric products, containing up to 135 needles [113]. NanoNC (Seoul, Korea) needle-based electrospinning, 3D electrospinning, and near-field electrospinning, with dual- or multi-channel syringe pumps, are available, and coaxial or tri-axial could be applicable [114]. Progene Link Sdn Bhd (Selangor, Malaysia) is widely used for producing nanofiber-based filters and face masks [115]. SKE (Research Equipment) (Bollate, Italy) systems offer various technological solutions to accurately control each process, guaranteeing batch-to-batch reproducibility and precise control of nanofiber parameters such as diameter, orientation and texture, co- and triaxial needles are also available [116]. Yflow^®^ fiber roller supplies more than one type of machine characterized by the following features: designed and optimized to work with electrospinning and electrospraying technologies; works with any polymer solution reported in the scientific literature with the continuous and non-stop operation. The machines are used in a wide variety of applications in diverse sectors, including the pharmaceutical industry [117]. Another example is ANSTCO (NF-LINE IV) (Tehran, Iran). The flexibility in design, ease of use, range of accessories provided, and high precision control and the wide range of variations considered for the main electrospinning parameters are the main outstanding features of ANSTCO electrospinning machinery. Hence, one can easily apply various electrospinning strategies and operational and environmental conditions to produce various nanofibrous products, especially nanofibrous air filters and wound dressing [118].

## 5. Electrospun Products for Commercial Purposes

The progress in electrospinning involves not only the machines and their accessories but also the electrospun products, so many companies have emerged in the last few years, offering electrospun commercial products that could be used for different applications (Table 2). Commercial usage of electrospun fibers across various fields is now possible.

### 5.1. Non-Medical Device (Non-MD) Products

Revolution Fibers Ltd. is New Zealand’s premier advanced materials company and a global leader in nanofiber production. Revolution Fibers Ltd. has commercialized products with various clients in the areas of filtration, skin health, composites, acoustics, biotech and anti-allergy bedding [143]. According to the data released by “Research and Markets”, the nanofibers’ global market can reach USD 1 billion by the end of 2021 [17].

Neotherix nanofiber scaffolds for tissue regeneration are bioresorbable scaffolds that possess a non-woven three-dimensional architecture, comprising nano/micro-scale synthetic bioresorbable polymer fibers. The highly porous scaffold structure supports the migration and proliferation of fibroblast cells from surrounding healthy skin tissue to facilitate wound healing [144]. Bio-Spun™: BioSurfaces’ patented electrospinning process allows for the incorporation of drugs, growth factors, radiopaque agents, or other bioactives directly into the fibers of both degradable and non-degradable materials with multiple advantages, such as high bioavailability, no extraneous binding agents are necessary, the ultra-high surface area provides a complete release of loaded drugs, and release rate can be tailored [145].

The Smart Mask filters out over 99%; it also offers virus trapping and bacteria killing and is highly comfortable due to its high breathability [146]. Reusable nanofiber-filtered masks use a nanofilter that maintains excellent filtering efficiency even after handwashing through the development of proprietary technology that aligns nanofibers with a diameter of 100~500 nm in orthogonal or unidirectional directions. This reusable nanofiltered face mask could relieve the challenges arising from the supply shortage of face masks [147]. The SETA nanofiber layer is a revolutionary approach to air filtration using electrospun nanofibers infused with antibacterial additives to trap even the smallest airborne parts, such as spores, allergens, and bacteria [148].

### 5.2. Medical Devices (MD) and Drug Delivery Systems (DDS)

ReBOSSIS is an innovative-type synthetic cotton-like bone-void/defect-filling material consisting of β-TCP (β-Tricalcium Phosphate), Bioabsorbable Polymer and SiV (Silicone-containing Calcium Carbonate that promotes the bone formation). Its cottony-type property makes it easier to handle at the time of operations compared to existing artificial bones. To give an example, unlike block-type solid artificial bone, processing can be done in advance to make ReBOSSIS fit into the shape or condition of different bone defects. Also, unlike granular-type artificial bone, ReBOSSIS does not fall from a bone-defect/void after filling. In addition to its good handling property, ReBOSSIS is featured with good elasticity and resilient capability, which is a great difference from the existing types of artificial bones. Being elastic and resilient, ReBOSSIS is designed to perfectly fill a bone void of any part of a patient’s body and in any size in a shorter time. Furthermore, ReBOSSIS can stay in a void firmly without any risk of falling from the void. Thereafter, ReBOSSIS replaces itself with the patient’s bone after healing [149]. Bioweb™ composites electrospun composites offer novel alternatives to traditional stent graft materials thanks to their microporous structure and increased surface area promoting cellular in-growth. Its high surface-to-weight ratio makes it suitable for tissue scaffolding. As a stent covering, the composites allow for the safe and non-inflammatory implantation of these devices [150]. HealSmart™ personalized wound care dressings are made from two types of microfiber polymers: hydrophilic (absorbing) and hydrophobic (moisture repelling). HealSmart™ is a patented microfiber technology that adapts to create dressings that are absorbent or hydrating, as specified. It contains polyhexamethylene biguanide (PHMB), an antimicrobial agent that protects healing cells from a bacterial proliferation in the dressing, and hyaluronic acid (HA), a naturally occurring glycosaminoglycan distributed widely throughout connective, epithelial, and neural tissues known to assist in wound healing. These dressing are FDA approved and have been clinically evaluated in over 12000 placements with a 96% satisfaction rate due to personalized treatment. Several clinical studies have demonstrated improvements in wound area reduction. Furthermore, the dressing composition is adjusted based on the wound’s functional needs. It eliminates variability and saves time, and the frequency of dressing changes can remain in place for up to 7 days [151]. In 2018, the Spain-based Bioinicia SL announced the approval for nanofiber-based drug delivery products, the Rivelin^®^ patch. The product is manufactured at Bioinicia’s industrial-scale nanofiber production facilities. It is a multi-component electrospun product designed for the unidirectional delivery of a pharmaceutical drug to a mucosal surface. Their controllable fiber diameter results in a large active-surface area and highly efficient drug release in the sub-micrometre range. Bioactive compounds are dissolved in the fiber matrix, effectively forming a uniform solid molecular solution that ensures high bioavailability even for poorly water-soluble substances. The wide range of materials that can be processed, including food- and pharmaceutical-grade biopolymers, synthetic polymers, and inorganic materials, allows for various drug release profiles, which can be tailored to a specific application [105]. The SWASA Surgical Mask is a three-layer membrane technology that protects from bacteria and other pollutions, UV sterile and anti-germ packing comfortable breathing [152]. AVflo™ is a superior alternative when considering vascular access for hemodialysis. Rapid hemostasis occurs in less than three minutes, no bleeding from the suture line, no weeping, rapid tissue integration, and no hematoma, seroma or pseudoaneurysm formation [153]. The PK Papyrus^®^ covered coronary stent system achieves greater bending flexibility and a smaller crossing profile than the traditional sandwich design stent. Electrospun polyurethane fibers on the stent surface create a thin and highly elastic membrane to seal perforations with high confidence [154]. The SurgiCLOT^®^ fibrin sealant patch is the first and only fibrin sealant designed specifically for bone bleeding, utilizing the dextran nanofibers to deliver a bolus of human fibrinogen and thrombin, augmenting the clotting cascade to promote a fast, strong and natural fibrin clot to aid the bone healing process [155].

### 5.3. D Cell Culture for Drug Development and Sensitivity Screening

It has been clear that three-dimensional (3D) cell culture offers a more realistic one step similar to the in vivo cell growth environment for the manipulation of cell properties than animals, and provides an alternative system to investigate the drug screening in the pre-clinical study phase [156] or to investigate the sensitivity of cancer cells to anticancer drugs [157]; therefore, the design and fabrication of a suitable 3D cell culture platform became an integral and indispensable part of the drug development process. As in other fields, some companies have succeeded in making excellent models for cell culture using the electrospinning process and have been incorporated with other commercial products, for example, Dipole Materials launched BioPaper^TM^Technology to help researchers improve pharmaceutical drug screening capabilities. The novel BioPaper technology is a 3D fibrous scaffold that easily fits all of the needs for 3D cell culture; it is biological-derived materials (including gelatin and collagen) with controlled properties and is easily handled for applications in high-throughput drug screening [158]. Another example of 3D cell culture is: nanofiber solution products (plates, chamber slides, dishes, and plate inserts from Nanofiber solutions). Nanofiber Solutions’ products are true 3D cell culture surfaces, optically transparent, compatible with light/visible microscopy, the physical dimensions are compatible with high-throughput and standard lab equipment, the fibers are made of polycaprolactone (PCL), the diameter of the nanofiber polymers is 700 nm, no special media or reagents are needed, and they are characterized by batch to batch consistency [159].

## 6. Future Perspective of Electrospinning and the Scale-up Production

Despite the revolution concerning scaling up instruments and products, it is documented that Nanospider is the only commercialized instrument used in pharmaceutical applications. Nevertheless, the research covering scaled-up ES technologies showed promising results and is expected to help the pharmaceutical industry’s field flourish. Therefore, industrial translation requires further developments and support from companies since most released studies are of academic origin. Based on nanofibrous systems’ morphological and physicochemical properties, it may become part of real pharmaceutical preparations in the future by increasing the solubility and dissolution rate of poorly soluble drugs [166], reducing the appropriate dose or repositioning existing chemical entities. It may be possible to use the fibrous structure for wound healing, absorbable dressings and topical preparations, grafts, stents, sclerotherapy balloons and diagnostic sensors, taking advantage of its stimulating effect on tissue regeneration [167]. There is also an example in the literature that preclinical-level, specific cell culture-based screening of drugs via nanofibers has been implemented [168,169]. According to the research considerations, the nanofibers can be used directly (e.g., buccal inserts or topical formulations) or with further processing as an intermediate to the final dosage form (e.g., tablets, capsules). In the drug-polymer nanofiber matrix, the drug remains in a higher energy amorphous state. It forms a well or improved soluble amorphous solid dispersion with the excipients used for formulation [170,171]. Although we find examples of extrusion-based formulations on the market [172], a fiber-based pharmaceutical formulation is not yet commercially available, since their thermodynamic stability and the prediction of amorphous–solid conversion needs more in-depth tracking. In both laboratory-scale and scale-up production, it is an essential criterion that the active substance is homogeneously distributed after incorporation into the fiber, remains stable in the preparation and meets strict regulatory requirements. The percentage of active ingredient incorporated into the fibers is low, up to ~10–20%. If the desired dosage form is a tablet, for a 200 mg API dose, the amount of fibrous phase product would be 1 g, which would result in a tablet of a size that is difficult for the patient to swallow. As noted above, nanofiber drug delivery systems in tablets or capsules may only be relevant for low-dose (1–1–10 mg) drugs, which leads to homogeneity. A promising alternative to the pharmaceutical application could be electrospinning for the gentle drying of bioactive substances [36]. A further limitation of the introduction of electrospinning into the pharmaceutical industry and its use in the preparation of a classical dosage form may be that the fibrous phase product cannot be subjected to further solution processes to preserve any amorphous solid dispersion so that the final dosage form can be formed for dry operations (e.g., compaction and homogenization), or limited to wet suspension operations. In the case of poorly soluble active ingredients, solvents may not be acceptable from official and patient safety. The production of medical or pharmaceutical medical devices and in-vitro diagnostics are promising areas of application for nanofiber systems where the thickness and orientation [173], chemical quality, and biodegradability [174] of the fibers can be finely tuned [175]. An example is a group of implants that help tissue regeneration, where nanometer-diameter, extremely long, individual fibers can help the fibroblast cells of a similar size range grow and adhere [176,177]. No study is available to discuss the effects of the scale-up of electrospinning for pharmaceutical purposes from various points of physical, physicochemical parameters, drug homogeneity, formulation stability, and manufacturing conditions, and robust manufacturing technology. In view of the successful achievements of the Bioinicia Company in the pharmaceutical industry, there is no longer an obstacle for the scaling-up of electrospinning, and it is only a time factor for further developments in the field of the pharmaceutical industry.

## 7. Conclusions

Electrospinning has been one of the innovative and potential methods for nanofiber (NF) production, especially for drug delivery systems. It is evident that nanofibers have great advantages and unique properties that enable the utilization of NF mats in a different area of drug delivery; for instance, the NFs produced by this method result in amorphous solid for poorly water-soluble drugs, making it a viable alternative to freeze drying and suitable for sensitive biopharmaceuticals. Furthermore, the proven possibilities for the downstream process of the fibers to prepare suitable dosage form necessitate mass production. The majority of the developed methods are based on increasing the jet numbers through: using multiple needles, needleless electrospinning, centrifugal spinning, hybrid electrospinning, and alternating current electrospinning. It has been demonstrated that free-surface, particularly spider spinning, and centrifugal methods exhibit high potential for high-throughput production rates, and centrifugal spinning has developed to be an efficient alternative to ordinary electrostatic-force-based spinning. Several companies offer specialized electrospinning equipment as well as nanofiber-based products for different applications, but the fewer are for the pharmaceutical industry. Electrospinning will remain a popular nanotechnology in laboratories, and the market of electrospinning equipment, both for laboratory research and for industrial production, is expected to grow significantly.

## Figures and Tables

**Table 1 pharmaceutics-13-00286-t001:** Summary of electrospinning setups by the technology used instrument scale and manufacturer.

Manufacturer (Country of Origin)	Setup Scales Available	Operation Capacities	Way of Properties	References
Electroblowing				
4Spin (Dolní Dobrouč, Czech Republic)	Laboratory	Unit type/Project system	Highly modular systems; Polysaccharides like Hyaluronic acid, Chitosan or Cellulose can be electrospun Fiber diameter: (>100) 300–500 nm	[98,119]
Needle/Nozzleless electrospinning				
Elmarco (Liberec, Czech Republic)	Laboratory and Industrial (Nanospider production line)	Unit type/Project system Mass production system Continuous system	Lab-based units: high throughput compared to needle-based systems; Industrial unit: low solvent consumption (minimalised usage and evaporation); Maximum effective nanofiber width: 1.6 m Fiber diameter: ● Industrial unit: ~150 nm +/− 30% SD; ● Mass production unit: 80–700 nm +/− 30% SD	[108,120,121]
SKE (Research Equipment) (Bollate, Italy) (by Elmarco) (Czech Republic)	Laboratory and Pilot and Industrial	Unit type/Project system Batch type systems	Also needle-based units available (laboratory scale); Pilot- and industrial-scale units are needleless Co- and triaxial needles available	[116,122]
Needle/Nozzle-based electrospinning				
Electrospinning (Tong Li Tech)/NaBond (Hong Kong)	Laboratory and Pilot and Industrial	Unit type/Project system Batch type systems Mass production systems	Needle-based electrospinning; Near-field electrospinning; Coaxial electrospinning; 3D print electrospinning; Melt electrospinning; Continuous electrospinning; For Scale-up units: 0.5–1–1.6–2 m width machines available; Basic, Professional, Scale-up, Robotic, All-in-one, Bio-medical, Portable instruments; Accessories: Multi-needle spinneret, tubeless spinneret, rotating and magnetic electrode collector, x-y moving stage, syringe and continuous pump;	[110,123]
Spraybase (Kildare, Ireland)	Laboratory	Unit type/Project system Batch type systems	Needle/Nozzle-based electrospinning; Melt electrospinning *; 5–40 um * (melt electrospinning); Coaxial/traixial kits available; 25–250 °C—PCL, PLA, PLGA, PP, PE, PMMA can be used (melt electrospinning)	[106,124]
Bioinicia (Valencia, Spain)	Industrial *	Mass production system Continuous system	* Electrospinning contractor; GMP and ISO validated for pharma and biomedical products.	[105,125]
E-Spin NanoTech Pvt. Ltd. (Uttar Pradesh, India)	Laboratory and Pilot	Unit type/Project system Batch type systems	Vertical, Horizontal, Inert Gas Spinning, Under solvent spinning; Ultra compact spinning chamber for low vacuum and inert gas spinning	[126,127]
Erich Huber GmbH (Gernlinden, Germany)	Laboratory	Unit type/Project system	Programmable spinning nozzle; Rotating collector/x-y-plate; 3D spinning possible Insulated, easy-to-clean system hood; 3D spinning movement speed: 1–600 mm/min	[99,128]
Fnm Co. (Fanavaran Nano-Meghyas) (Baghestan, Iran)	Laboratory and Pilot and Industrial	Unit type/Project system Batch type systems Mass production systems Continuous system	Lab-scale unit: Dual Pump Electrospinning Machine (Side by Side Electroris^®^) For polymeric/carbon/ceramic nanofibers with diameter range of 50 nm to a few micron core-shell nanofibers. One–eight spinning units for industrial unit, all parameters can be set separately Fiber diameter: 60–500 nm	[109,129]
Fluidnatek (by Bioinicia) (Valencia, Spain)	Laboratory and Pilot	Unit type/Project system Batch type systems	Large-volume solution feeding system for extended production batches (single-phase or coaxial)* Multihead emitter system with larger-volume solution reservoirs; Solvent resistant housing; Easy-to-clean contruction for most models—ideal for cleanrooms; Package for GMP validation available (or ISO13485)	[105,125]
HOLMARC Opto-Mechatronics (Kerala, India)	Laboratory	Unit type/Project system Batch type systems	Protein nanofibers, carbon nanotubbes, inorganic nanofibers; UV curing lamp (254 nm) can be added on the top of the rotating collector to cure the spun fibers; Holmarc’s model (HO-NFES-SYS): Nano Fiber Double Spinning and Yarning system Fiber diameter: 50–5000 nm	[100,130]
INOVENSO (İstanbul, Turkey)	Laboratory and Pilot and Industrial	Unit type/Project system Batch type systems Mass production systems Continuous system	Single- and multi-nozzle systems for lab-scale workflow available; Maximum number of nozzles: 204 (industrial-scale machine); 180–5000 m^2^/day, 5000 g nanofiber/day. 50–400 nm diameter; 49,350 m^2^/day (23.5 h/day) Fiber diameter: 50–400 nm diameter	[111,131]
KatoTech Co. Ltd. (Kyoto, Japan)	Laboratory	Unit type/Project system	This device is widely used in the automotive industry for research and development of filters and fuel cells. Fiber diameter: 50–800 nm	[65,101]
LINARI NanoTech (Pisa, Italy)	Laboratory and Industrial	Unit type/Project system Batch type systems	Coaxial needle/multineedle systems; Up to eight independently controlled syringe pumps; Automatic cleaning of needles. Internal temp. and humidity control.	[112,132]
MECC Co. Ltd. (Fukuoka, Japan)	Laboratory and Pilot	Unit type/Project system Batch type systems	Dedicated device for healthcare/medical application; Production of nonwoven nanofiber (effective width: 0.4–1 m); Fiber diameter: 10 nm- several micrometers	[133,134]
Nadetech Innovations (Navarra, Spain)	Laboratory	Unit type/Project system	Spinnerets: single/coaxial/traixial/multi- nozzle	[102,135]
Nanoflux (Singapore)	Laboratory and Pilot and Industrial	Unit type/Project system Batch type systems Continuous system	Multi-nozzle system for the continuous production of the nanofiber fabric products; Up to 135 needles; High-temperature unit (up to 280 °C available)	[113,136]
NanoNC (Seoul, Korea)	Laboratory and Pilot and Industrial	Unit type/Project system Batch type systems Mass production systems Continuous system	Dual/multi-channel syringe pumps available; Coaxial/triaxial/precision/multi/heating/micro nozzle options; (Rotary jet and wet spinning machines available)	[114,137]
Physics Equipment (Chennai, India)	Laboratory	Unit type/Project system Batch type systems	Spinning Chamber: Constructed with Aluminium Strut Frames. ● Two Polycarbonate- Clear doors ● Window on one side ● Fiberglass Panel on other sides ● With Sensor to switch of H.V.Power Supply	[103,138]
Progene Link Sdn Bhd (Selangor, Malaysia)	Laboratory and Pilot and Industrial	Unit type/Project system Batch type systems Mass production systems Continuous system	Industrial unit: In each unit, there is a rotating drum dipped in the polymeric solution, and a plate/rotating drum collector placed on the top of each unit.	[115,139]
Spinbox (by Bioinicia) (Valencia, Spain)	Laboratory	Unit type/Project system Batch type systems	Basic/Intermediate/Advanced kits available; Spare parts available; Engineered by Bioinicia and Fluidnatek systems, for research purposes	[105,125]
SPINBOW (San Giorgio di Piano Italy)	Laboratory and Pilot	Unit type/Project system	Feeding unit with infusion pump (up to four syringes); Linear sliding system with a reciprocating motion housing (up to four needles) spinneret Rotating interchangeable drum collector	[140,141]
Yflow (Málaga, Spain)	Laboratory and Pilot and Industrial	Unit type/Project system Batch type systems Mass production systems Continuous system	Upgrade: additional syringe pump for coaxial spinning; coaxial nozzle injector, Taylor cone visualization system. (From Professional system upwards upgrades are included)	[117,142]

* refers to the industrial electrospinning contractor indicated in the second coloumn.

**Table 2 pharmaceutics-13-00286-t002:** Summary of electrospun products by product grade. MD = Medical Device, DDS = Drug Delivery System

Product Type (e.g., Technology, Device, Filter, Mask, etc.)	Brand Name *(Type)*	Product Grade	Manufacturer	Detailed Specification (Availability, Key Features, etc.)	References
*Publicly available products*					
Face mask *(nanofiber for mask or filter technology)*	SWASA^®^ face mask (odorless, plus) N95/N99 Face mask technology	MD *(recommended for medical doctors)*	E-SPin NanoTech Pvt. Ltd. (Uttar Pradesh, India)	Patented face mask technology	[152]
	SWASA^®^ Surgical mask	MD		Surgical face mask	
Surgical implants and wound treating products/wound dressings	AVflo™	MD	Nicast (Lod, HaMerkaz, Israel)	Unique Nanofibrous Vascular Access Graft	[153]
	PK Papyrus^®^		Biotronic (Berlin, Germany)	Electrospun polyurethane fibers on stent surface; thin and highly elastic membrane	[154]
	Surgiclot^®^		St. Theresa Medical Inc. (Eagan, USA)	Dextran nanofibers; fibrin sealant designed specifically for bone bleeding;	[155]
	NanoCare^®^		Nanofiber Solutions™ (Ohio, USA)	Veterinary product; ECM-like fiber structure	[160]
	Phoenix Wound Matrix RenovoDerm^®^			Treatment of both partial- and full-thickness wounds	[161]
	Zeus Bioweb™		Zeus Industrial Products, Inc. (Orangeburg SC, USA)	Ultrasmall PTFE polymeric fibers with low chemical reactivity	[150]
	ReBOSSIS^®^		Ortho ReBirth (Yokohama-shi Kanagawa pref., Japan)	Bone-void/defect-filling material; Components: TCP (β-Tricalcium Phosphate), Bioabsorbable Polymer and SiV (Silicone-containing Calcium Carbonate that promotes the bone formation).	[149]
	ReDura™		MEDPRIN (Guangzhou, China)	FDA approved degradable material poly-L-lactic acid (PLA) Similar to native extracellular matrix (ECM), rapid repair and regeneration.	[162]
	HealSmart™	DDS	PolyRemedy^®^, Inc. (Concord, MA, USA)	Antimicrobial Dressings with the addition of Hyaluronic Acid (HA)	[151]
	3D Insert^TM^-PCL	DDS	3D Biotek (New Jersey, USA)	Biodegradable polyester material that has been used in many FDA approved implants, drug delivery devices, suture, adhesion barrier.	[163]
*Products under development*					
Patches	Pathon	other	N/A	N/A nitric oxide releasing patch	[164]
	Rivelin^®^ patch	DDS	Bioinicia (Valencia, Spain)	Designed for unidirectional drug delivery to a mucosal surface.	[165]

## Data Availability

Not applicable.

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
