# Peer review of "Scale-up of Electrospinning: Market Overview of Products and Devices for Pharmaceutical and Biomedical Purposes"

_pharmaceutics, 2021, doi:10.3390/pharmaceutics13020286_

Round 1
Reviewer 1 Report
The manuscript, according to the title, aims to clearly describe the state of knowledge of the electrostatic spinning method with higher production of nanofibres for applications in the pharmaceutical industry. The text is divided into five chapters, while the Introduction section does not offer an explanation of the major obstacles faced by the pharmaceutical industry in manufacturing nanofibre products. The reader learns the generally known facts of the results at a level of laboratory research that does not directly focus on the intended area. Similarly, in the next part, the individual paragraphs address the general procedures for increasing the capacity of this method, resulting in no conclusions applicable specifically to the pharmaceutical industry. It is precisely the industry that faces specific obstacles with electrospinning that the authors ignore here. The third part, devoted to equipment, is not again focused on the said industry, besides that contains many ambiguities and shortcomings. Furthermore, the paragraph draws on the device manufacturers' websites, neither on the scientific nor patent literature. Current commercial products are listed with no information expected or required by the readers from the field. The text does not tell us about the composition of the products, the properties and importance of nanofibre use at all. Overall, the manuscript is its content at the level of a student research work, not a review of scientific publication. For this reason, I cannot recommend the manuscript for publication.
Author Response
The manuscript, according to the title, aims to clearly describe the state of knowledge of the electrostatic spinning method with higher production of nanofibres for applications in the pharmaceutical industry. The text is divided into five chapters, while the Introduction section does not offer an explanation of the major obstacles faced by the pharmaceutical industry in manufacturing nanofibre products.
We agree with the comment and a separate paragraph was included in the revised paper about the major obstacles faced by the pharmaceutical industry in manufacturing electrospun nanofiber products.
The reader learns the generally known facts of the results at a level of laboratory research that does not directly focus on the intended area. Similarly, in the next part, the individual paragraphs address the general procedures for increasing the capacity of this method, resulting in no conclusions applicable specifically to the pharmaceutical industry. It is precisely the industry that faces specific obstacles with electrospinning that the authors ignore here.
The authors reworked the concerned part of the manuscript and provided the specific obstacles of the industrial implementation of electrospinning.
The third part, devoted to equipment, is not again focused on the said industry, besides that contains many ambiguities and shortcomings. Furthermore, the paragraph draws on the device manufacturers' websites, neither on the scientific nor patent literature. Current commercial products are listed with no information expected or required by the readers from the field. The text does not tell us about the composition of the products, the properties and importance of nanofibre use at all.
The authors revised that part of the paper and the Table including the equipment was modified. The capacity of the equipment was provided along with the pharmaceutical application areas indicating the scientific literature of the patent.
Overall, the manuscript is its content at the level of a student research work, not a review of scientific publication. For this reason, I cannot recommend the manuscript for publication.
The manuscript was substantially reworked and the authors hope that the revised version meets the Reviewer’ expectations.
Reviewer 2 Report
The manuscript describes the up-scaling of nanofibrous webs' production. It introduces electrospinning and centrifugal spinning and then briefly describes up-scaling challenges, electrospinning machines available on the market, and electrospun products used for commercial purposes. The title suggests that the manuscript describes more details essential for the pharmaceutical industry. However, the manuscript sounds more for general readers due to incorporating many examples of medical devices and lab consumables.
The manuscript should describe nanofibrous drug delivery devices and challenges when up-scaling from the lab or pilot scale. Are there any examples illustrating differences in physical properties of as-spun fibers prepared with lab and industrial-scale machines for the same polymer solution? This is crucial in the context of drug release patterns that can be significantly different for both materials.
Please provide information about the machines in the context of their operational capabilities, which of them work in a continuous way and batch etc.
Another aspect that I would consider is cell culture for in vitro assays in drug screening etc. Are there any examples present in the scientific literature?
The last issue refers to future perspectives mentioned in the title. There should be a separate paragraph about it because it is impossible to find anything about this topic in the manuscript in the present form.
In general, after applying the corrections, the paper can be of significant value for readers. However, many typos and grammar corrections are needed before publication.
Author Response
The manuscript describes the up-scaling of nanofibrous webs' production. It introduces electrospinning and centrifugal spinning and then briefly describes up-scaling challenges, electrospinning machines available on the market, and electrospun products used for commercial purposes. The title suggests that the manuscript describes more details essential for the pharmaceutical industry. However, the manuscript sounds more for general readers due to incorporating many examples of medical devices and lab consumables.
We agree with the comment and rephrase the manuscript focusing on the pharmaceutical industrial applications.
The manuscript should describe nanofibrous drug delivery devices and challenges when up-scaling from the lab or pilot scale. Are there any examples illustrating differences in physical properties of as-spun fibers prepared with lab and industrial-scale machines for the same polymer solution? This is crucial in the context of drug release patterns that can be significantly different for both materials.
We added a paragraph about the comparison of the functionality-related characteristics of laboratory-scale and industrial-scale nanofibers.
Please provide information about the machines in the context of their operational capabilities, which of them work in a continuous way and batch etc.
In the revised version the authors provided additional information about the operational capacities and the way of preparations, which were also included in Table 1.
Another aspect that I would consider is cell culture for in vitro assays in drug screening etc. Are there any examples present in the scientific literature?
The preparation of electrospun cell culture would be of impact from the point of the preclinical screening of various drugs. As such it would be particular interest in the in vitro assays of great translational potential. This aspect of potential application was also included in the revised version of the paper.
The last issue refers to future perspectives mentioned in the title. There should be a separate paragraph about it because it is impossible to find anything about this topic in the manuscript in the present form.
The authors added a separate paragraph about the future perspectives of scaled-up electrospinning.
In general, after applying the corrections, the paper can be of significant value for readers. However, many typos and grammar corrections are needed before publication.
The revised version of the manuscript was language-edited.

Reviewer 3 Report
It is not true that electrospinning is "used for the synthesis of nanomaterials since 19th century [16]" (see line 38). First patent in electrospinning was published in 1900 by Cooley J.F. (Patent GB 06385 “Improved methods of and apparatus for electrically separating the relatively volatile liquid component from the component of relatively fixed substances of composite fluids” 19th May 1900). So, it was early beginning of 20th century. Works of Strutt J.F. (Rayleigh Lord), published in 1878-1882, are a bit related to electrospinning but was not about the synthesis of fibres. More about history of electrospinning authors can found in a journals related to the textile as nanofibres are a part of textile materials (for example DOI: 10.1515/aut-2018-0021).
In review papers authors need cite the originals of papers in which one or other phenomenon was described at first. If authors cite G. Taylor (see line 43), they need to add to the list of References his work, not only works in which Taylor's works are cited or in which are written about G. Taylor works.
An error in line 245 - SKE (Research Equipment) is Italian company, while Elmarco is not Italian company, it is company of Czechia.
Author Response
It is not true that electrospinning is "used for the synthesis of nanomaterials since 19th century [16]" (see line 38). First patent in electrospinning was published in 1900 by Cooley J.F. (Patent GB 06385 “Improved methods of and apparatus for electrically separating the relatively volatile liquid component from the component of relatively fixed substances of composite fluids” 19th May 1900). So, it was early beginning of 20th century. Works of Strutt J.F. (Rayleigh Lord), published in 1878-1882, are a bit related to electrospinning but was not about the synthesis of fibres. More about history of electrospinning authors can found in a journals related to the textile as nanofibres are a part of textile materials (for example DOI: 10.1515/aut-2018-0021).
We fully agree with the comment and modified the paper accordingly.
In review papers authors need cite the originals of papers in which one or other phenomenon was described at first. If authors cite G. Taylor (see line 43), they need to add to the list of References his work, not only works in which Taylor's works are cited or in which are written about G. Taylor works.
We added the original references of Taylor’s work in the revised version of the paper.
An error in line 245 - SKE (Research Equipment) is Italian company, while Elmarco is not Italian company, it is company of Czechia.
It was our mistake, and corrected it in the revised version of the paper.

Round 2
Reviewer 1 Report
I appreciate the addition and extension of the text to respond to the comments made. In this way, the manuscript can be considered complete and finished.
Unfortunately, as an opponent, I am not convinced that the reader will find in this publication what he would have expected from the title. Who is the article for and what new information does it provide? If a reader needs to produce nanomaterials for medical devices, he or she will not know what technology needs to meet and which ones to prioritize when choosing from the list. If a reader develops such technologies, they will not know how to design the device for the stated purposes and industry. The reader will not know how the pharmaceutical industry assesses the quality of nanofibers, how to increase or meet it (see rows 61, 72, 180, 431, 625, etc.). The manuscript does not in any way reflect or suggest solutions to the real problems that are addressed in the scale-up electrostatic spinning method in the pharmaceutical industry (there are many, and the text omits them). The article simply describes what has already been done and does not clearly capture the essence of the current situation in the sector. For this reason, too, I am not proposing specific changes, because I think that an article with such ambition and title should be handled and focused significantly differently. These are the shortcomings that prevent me from agreeing to publish the manuscript in such a valuable journal.
Author Response
Dear Reviewer,
The authors greatly appreciate the Reviewer's comments and agree that the article describes what has already been done in the pharmaceutical and biomedical sector using scale-up electrospinning technology. However, the available products are optimised for their functionality-related characteristics, consequently, only limited literature is available about the correlation between the process parameters and the quality attributes of the final fibre-based products. Since the reviewer did not suggest any concrete correction, the authors decided to change the title of the paper, which matches the content better. I hope now the paper meets the expectations.
Reviewer 2 Report
The manuscript describes the up-scaling of nanofibrous webs' production. It introduces electrospinning and centrifugal spinning and then briefly describes up-scaling challenges, electrospinning machines available on the market, and electrospun products used for commercial purposes. As it was mentioned in previous Review Report, the title suggests that the manuscript describes more details essential for the pharmaceutical industry. Some of the concerns have been addressed. Going back to the other comments, I believe that all of them were addressed and in overall the manuscript was improved.
Author Response
Dear Reviewer,
Thank you for the positive comments about the revised version of the paper.
According to Reviewer 1' remarks, the authors decided to change the paper's title to the following, which better matches the content.
The modified title is the following:
Scale-up of Electrospinning: Market overview of products and devices for pharmaceutical and biomedical purposes
Round 3
Reviewer 1 Report
Accept in present form.